# Intrinsic Memory Agents: Heterogeneous Multi-Agent LLM Systems through Structured Contextual Memory

## Abstract

Multi-agent systems built on Large Language Models (LLMs) show exceptional promise for complex collaborative problem-solving, yet they face fundamental challenges stemming from context window limitations that impair memory consistency, role adherence, and procedural integrity. This paper introduces Intrinsic Memory Agents, a novel framework that addresses these limitations through agent-specific memories that evolve intrinsically with agent outputs. Specifically, our method maintains role-aligned memory that preserves specialized perspectives while focusing on task-relevant information. Our approach utilises a generic memory template applicable to new problems without the need to hand-craft specific memory prompts. We benchmark our approach on the PDDL, FEVER, and ALFWorld datasets, comparing its performance to existing state-of-the-art multi-agentic memory approaches and showing state-of-the-art or comparable performance across all three, with the highest consistency. An additional evaluation is performed on a complex data pipeline design task, and we demonstrate that our approach produces higher quality designs across 5 metrics: scalability, reliability, usability, cost-effectiveness, and documentation, plus additional qualitative evidence of the improvements. Our findings suggest that addressing memory limitations through intrinsic approaches can improve the capabilities of multi-agent LLM systems on structured planning tasks.

## 1 Introduction

Recent advances in large language models (LLMs) have enabled their application as autonomous or semi-autonomous agents capable of complex reasoning and decision-making (Huang et al., 2024). Multi-agent LLM systems, where multiple LLM instances interact to solve problems collaboratively, have shown particular promise for tasks requiring diverse expertise (Park et al., 2023; Qian et al., 2025). These systems leverage the complementary capabilities of specialized agents to address challenges that would be difficult for single-agent approaches to resolve effectively.

Despite their theoretical advantages, multi-agent LLM systems face several implementation challenges that limit their practical effectiveness, from coordination overhead, to the consistency in role adherence among the agents (Li et al., 2024c). Most critically, the fixed-size context windows of LLMs restrict their ability to maintain long-term conversational context, an issue that is exacerbated in multi-agent frameworks with multiple agents in a single conversation. This leads to issues such as perspective inconsistency, forgetting key requirements, and procedural drift. Current solutions such as Retrieval-Augmented Generation (RAG) (Lewis et al., 2020; Gao et al., 2024) and agentic memory approaches (Packer et al., 2024; Xu et al., 2025; Chhikara et al., 2025) are designed for single-agent and user interaction scenarios, which do not account for the volume of information growing with the number of agents.

To address these challenges, we introduce Intrinsic Memory Agents, a novel multi-agent architecture that uses agent-specific memories aligned with conversational objectives. Unlike previous approaches, our system updates memories that are specific to each agent, ensuring heterogeneity and memories that reflect both historical context and recent developments while preserving agent-specific perspectives. The intrinsic nature of memory updates, derived directly from agent outputs

rather than external summarization, ensures unique memories that maintain consistency with agent-specific reasoning patterns and domain expertise. We evaluate our approach through benchmarking and through a specific data pipeline design case study to show its practical usage. The evaluation demonstrates that our Intrinsic Memory Agents approach yields significant improvements in conversational coherence, role consistency, and collaborative efficiency compared to conventional multi-agent implementations. These improvements translate to qualitative enhancements in solution quality without increasing the number of conversation turns, suggesting broad applicability across domains where multi-agent LLM systems are deployed.

The main contributions of our work are as follows:

- **Intrinsic Memory Updates**: Memory updates derived from agent outputs rather than external summarization.
- **Agent-Specific Memory**: Independent memories maintained for each agent to preserve perspective autonomy.

## 2 RELATED WORK

Recent years have seen significant progress in the development of multi-agent systems powered by LLMs. These systems have been applied in various domains, such as software development, scientific experimentation, gaming, and social simulation (Li et al., 2024c). For example, in software development, multi-agent systems enable concurrent consideration of architectural design, security, user experience, and performance optimization (Hong et al., 2024). Hallucinations due to outdated knowledge or retrieval extraction issues remains a major challenge which limits the effectiveness of multi-agent systems Huang et al. (2025). The use of a shared knowledge base or memory storage is an important aspect to maintain up-to-date, coherent and correct information among agents.

### 2.1 MEMORY IN AGENT-BASED SYSTEMS

In agent-based systems, memory is pivotal for maintaining context, learning from historical interactions, and making informed decisions. As Zhang et al. (2024) noted, memory supports tasks such as ensuring conversation consistency and effective role-playing for single-agent systems. In multi-agent systems, memory facilitates coordination, communication, and collaborative problem-solving, as Guo et al. (2024) discussed.

Memory in LLMs can be categorized under short-term memory and long-term memory. Short-term memory is information that fits within the model's fixed context window. Commercial LLMs such as GPT-4o (OpenAI, 2024) and Claude (Anthropic, 2024) are able to process large contexts of over 100K tokens, with some models such as Gemini 2.5 Pro (Comanici et al., 2025) able to process over 1 million tokens in its context window. However, the hard limit of the context window size remains, and increasing the context length does not necessarily increase reasoning or learning capabilities of the LLM (Li et al., 2024b). This is because the long context can move the relevant information further away from each other in the context window.

Long-term memory is information that persists beyond the context window or single instance of an LLM. This information can be stored in external databases and retrieved using RAG techniques (Lewis et al., 2020; Gao et al., 2024). Long-term memory aims to alleviate the issue of short-term memory's limited capacity, but introduces other disadvantages such as retrieval noise, the complexity of building a retrieval system, latency, and storage costs (Asai et al., 2024; Yu et al., 2024).

The limitations of context length and existing memory mechanisms are particularly pronounced in multi-agent settings, where the volume of information exchanged grows with the number of agents involved Li et al. (2024a). As multi-agent conversations extend, the probability of critical information being at a long distance or even excluded from the accessible context increases dramatically. This information loss undermines the primary advantage of multi-agent systems: The integration of diverse, specialized perspectives toward cohesive solutions He et al. (2025). This is exacerbated by current long-term memory approaches which provide a homogeneous memory for the agents, decreasing the benefits of having agents focused on a single part of the task. Our proposed approach therefore focuses on the heterogeneity of agents and their memories, ensuring that each agent maintains a memory that is uniquely relevant to their role.

## 2.2 AGENTIC MEMORY

Agentic memory offers a solution to long-term memory and limited contextual information by periodically condensing conversation history into concise summaries (Wang et al., 2025; Chen et al., 2024). These approaches generate sequential or hierarchical summaries that capture key decisions and insights from previous exchanges. Some agentic memory approaches combine with RAG approaches by storing the summarized contexts for retrieval later in the conversation (Xu et al., 2022), or by storing in- and out-of-context memory in a hierarchical system to dynamically adapt the current context (Packer et al., 2024; Xu et al., 2025). While agentic memory methods provide better contextual integration than pure retrieval approaches, they frequently lose critical details during the condensation process. Furthermore, the undirected and unstructured nature of general summarization often fails to preserve role-specific perspectives and specialized knowledge that are essential to effective multi-agent collaboration.

Our proposed Intrinsic Memory Agents similarly uses an agentic memory approach to summarize and store information. Unlike existing approaches, we introduce heterogeneous memory for each agent in the multi-agent system to maintain specialized roles in collaborative tasks, and apply a templated approach to each agent ensuring cohesive memory throughout. This addresses the limitations of existing memory mechanisms by ensuring that each agent maintains its own memory, reflecting both historical context and new information while maintaining heterogeneous agent-specific perspectives and expertise.

## 3 INTRINSIC MEMORY AGENTS

The various agentic memory approaches are all designed in single-agent scenarios to remember crucial details when interacting with an end-user. Due to the multi-turn long conversations between agents, a direct implementation of single-agent agentic memory becomes complicated and resource-intensive, with each agent requiring retrieval systems and distinct contextual updates.

We propose Intrinsic Memory Agents, a framework for multi-agent LLM systems that maintains agent-specific memories aligned with conversational objectives. Figure 1 illustrates the architecture of our Intrinsic Memory Agents framework. In this approach a query is made by the user, the first agent makes a comment based on its role description, the conversation is updated, followed by a memory update for the agent that commented, there is a check for consensus and the cycle starts again. The context in this case is made up of both the agent's intrinsic memory and the conversation, meaning that as the conversation continues the agents increasingly diverge in their interpretation of that context.

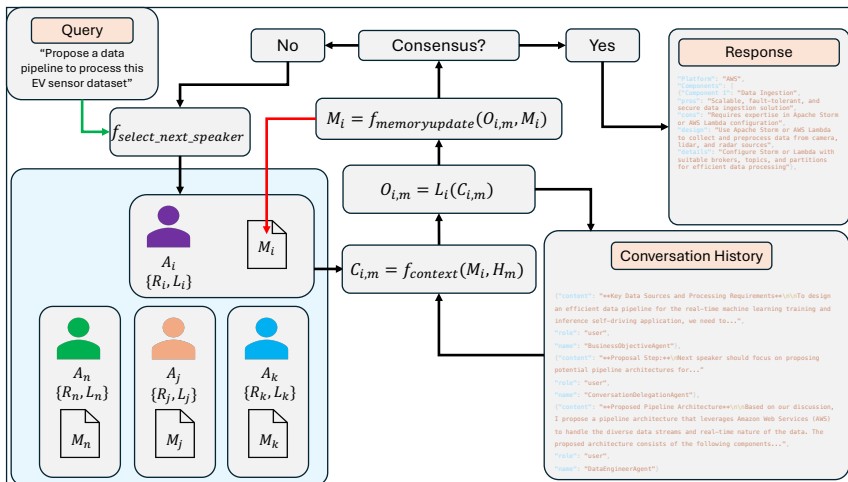

Figure 1: Intrinsic Memory Agents Framework. For $n$ agents and $m$ conversation turns, each agent $A_n$ contains its own role description $R_n$ and language model $L_n$. Its memory $M_{n,m}$ is updated based on the input context $C_{n,m}$ and output $O_{n,m}$.

## 3.1 Framework Definition

Let us define the multi-agent system $\mathcal{A} = \{A_1, A_2, ..., A_N\}$ consisting of $N$ agents. Each agent $A_n = \{R_n, M_n, LLM_n\}$ is characterized by a role specification $R_n$ that defines the agent's expertise domain and objectives, a memory $M_n$ that evolves throughout the conversation, and an LLM instance $LLM_n$, which may share parameters between agents.

The conversation consists of a sequence of turns $T = t_1, t_2, ..., t_M$ where each turn $t_m$ involves an agent selection function $\sigma(t_m) \to A_n$ that determines which agent speaks, an input context $C_{n,m}$ constructed for the selected agent, an output $O_{n,m}$ generated by the selected agent, and a memory update operation for the selected agent.

Critically, our framework separates the input context construction and memory update processes, allowing for agent-specific memory maintenance while preserving a shared conversation space.

## 3.2 Memory Update mechanism

For each agent in the system, we maintain a memory $M_n$ that evolves over time. Let $M_{n,m}$ represent the memory of agent $n$ after $m$ conversation turns. The memory update process works as follows:

Agent $A_n$ receives input context $C_{n,m}$ consisting of relevant conversation history $H_m$ and previous memory $M_{n,m-1}$,

$$C_{n,m} = f_{\text{context}}(H_m, M_{n,m-1}); \tag{1}$$

and agent $A_n$ generates output $O_{n,m}$ using the underlying LLM $L_n$,

$$O_{n,m} = L_n(C_{n,m}). \tag{2}$$

Then with the generated output $O_{n,m}$ and the previous memory $M_{n,m-1}$, we update the slot content using a memory update function,

$$M_{n,m} = f_{\text{memory\_update}}(M_{n,m-1}, O_{n,m}). \tag{3}$$

The memory update function $f_{\text{memory\_update}}$ is implemented as a prompted LLM operation. Specifically, for the previous memory $M_{n,m-1}$ at turn $m-1$ and agent output $O_{n,m}$ at turn $m$, the update function constructs the prompt as shown in Figure 8. The LLM's response to this prompt becomes the updated memory $M_{n,m}$. The context construction function $f_{\text{context}}$ presented in equation 1 determines what information is provided to an agent when generating a response. The algorithm takes the existing conversation history and agent memory, appending both to the context and using the remaining tokens to include the rest of the conversation history. The full algorithm pseudo-code is displayed in the Appendix A Algorithm 1.

This algorithm prioritizes:

1. The initial task description to maintain objective alignment.
2. The agent's structured memory to preserve role consistency.
3. The most recent conversation turns to maintain immediate context.

By prioritizing memory inclusion over exhaustive conversation history, the algorithm ensures that agents maintain role consistency and task alignment even when conversation length exceeds context window limitations. We conduct an ablation study on the structure of the memory template in appendix B. The ablation study shows the use of a generic or dynamic LLM-generated template shows consistently better performance compared to a hand-crafted template, which is prone to sensitivity if a poorly created template is used.

## 4 Quantitative benchmarks

To evaluate our approach, we test our memory agents against the PDDL (Planning Domain Definition Language), FEVER (Fact Extraction and VERification) Thorne et al. (2018), and ALFWorld Shridhar et al. (2021) numeric benchmarks. PDDL involves structured planning tasks from Agent-Board (Ma et al., 2024), where the agents generate executable plans for abstract problem domains,

evaluating their reasoning and coordination. FEVER is a dataset for evidence-based claim verification, requiring agents to retrieve and reason over textual evidence and assess a given factual claim. Finally, ALFWorld Shridhar et al. (2021) is a text-based interactive environment which simulates household tasks with natural language instructions and descriptions. It tests an agent's ability to navigate and execute complex sequential actions to complete tasks.

For numerical benchmarks, we follow the same experimental methodology as G-Memory (Zhang et al., 2025), another memory framework for multi-agent systems. We re-run the G-Memory framework [1] as we cannot directly compare to the published G-Memory results which were benchmarked with GPT-4o-mini as the base language model. We chose to use the G-Memory framework as a comparison as the framework implements a variety of existing memory architectures, allowing us to compare our Intrinsic Memory Agents with existing architectures and benchmarks. G-Memory uses Autogen for multi-agent simulation, matching our use of Autogen for our architecture. We chose to use the three benchmarks to cover a range of structured planning, comprehension, and reasoning tasks, all of which are aspects aligned with the data pipeline case study detailed in Section 5. We run Gemma3:12b for the numeric benchmarks using Ollama [2] with 5 independent runs, each with their own set seeds for reproducibility. We use a larger model for the numeric benchmarks as initial tests on the Llama3.1:3b model found poor results for every benchmark and memory framework. Our computational infrastructure utilizes a high performance computing cluster with A100 GPUs, running on GNU/Linux 4.18.0-553.el8_10.x86_64.

## 4.1 BENCHMARKING RESULTS

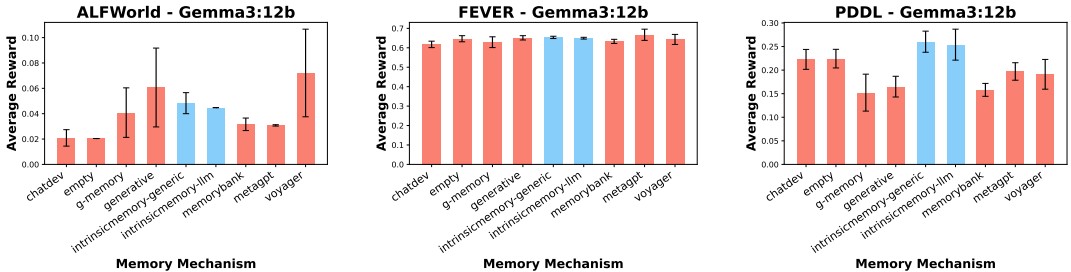

Figure 2: Intrinsic Memory performance across the three benchmarks, the blue bars are our Intrinsic Memory.

Figure 2 shows the average reward of each memory system benchmarked against ALFWorld, FEVER, and PDDL, the error bars showing the standard deviation across 5 independent runs. While our Intrinsic Memory mechanism doesn't obtain the highest rewards in each benchmark, there are results to indicate our approach has consistent strong performance compared to the other memory mechanisms.

In the ALFWorld benchmark, the Voyager and Generative approaches obtain the highest average reward, at 0.072 and 0.061 mean reward respectively. However, they also show the highest standard deviation among all memory mechanisms, at 0.035 and 0.031 respectively, indicating their high variability. In contrast, our method with both the generic and LLM-generated templates obtains the 3rd and 4th best performance, at 0.048 and 0.045 mean rewards, with much lower variance at 0.0083 and 0.0003 respectively. These results indicate our strong performance and consistency when solving the ALFWorld set of problems. Similarly for the FEVER dataset, all memory approaches obtain similar performance, with the best approach, MetaGPT, also showing the highest standard deviation compared to other approaches. Our Intrinsic Memory mechanism shows the lowest standard deviation on this dataset, with mean rewards ranked second for the generic template, and fourth for the LLM-generated template, showing more evidence of our consistency. Finally in the PDDL benchmark, both our Intrinsic Memory approaches outperform all other memory mechanisms, with not a significantly higher standard deviation than other approaches, at 0.260 and 0.254 mean rewards for the generic template and LLM-generated templates respectively. Table 4 in the appendix shows

---

[1] https://github.com/bingreeky/GMemory

[2] https://ollama.com/library

the mean rewards, standard deviation, and average token counts for each benchmark and memory mechanism in detail.

The PDDL dataset are structured planning tasks, which fits the intended use case of Intrinsic Memory for agent discussion, planning and design. As Intrinsic Memory assigns agent-specific memory, it can more clearly distinguish planning and actions to complete tasks. More tokens are used by Intrinsic Memory to generate structured templates per agent per round of discussion, and is a worthwhile trade-off in both reward score and token efficiency. In contrast, the FEVER dataset tasks are meant for fact extraction where reasoning plays a larger role than raw memory. We find that our Intrinsic Memory performs just as well as other memory methods, indicating memory methods in general are less applicable to the FEVER problems, and that our performance is in line with other memory mechanisms. Finally in ALFWorld, the two best performing memory mechanisms, Voyager and Generative, also have the highest standard deviation, showing a lack of consistency compared to the Intrinsic Memory approach, where the agent-specific memory helps maintain consistent performance compared to global or cross-trial memory implementations.

# 5 DATA PIPELINE DESIGN CASE STUDY

As a practical case study to evaluate our approach, we applied our memory agents to a collaborative data pipeline design, a complex task requiring multiple perspectives. We run 10 independent outputs with eight specialized agents:

1. **Evaluation Agent (EA)** evaluates the output solutions.

2. **Knowledge Integration Agent (KIA)** summarizes each discussion round (e.g. after every agent has contributed at least once).

3. **Data Engineer Agent (DEA)** determines the data processing needs.

4. **Infrastructure Engineer (IA)** designs the cloud infrastructure.

5. **Business Objective Engineer (BOA)** checks against business requirements.

6. **Machine Learning Engineer (MLE)** provides ML implementation.

7. **Conversation Delegation Agent (CDA)** is responsible for facilitating the collaborative process.

8. **Documentation Joining Agent (DJE)** is responsible for producing final output after consensus is reached among agents.

The agents are tasked with designing a cloud-based data pipeline architecture through a structured process involving proposals, discussions, and consensus formation. The full prompts and task descriptions can be found in Appendix D.

The output requirements include a concise summary, high-level plan, resource estimates, and a structured JSON specification.

## 5.1 SYSTEM CONFIGURATIONS

We evaluated two system configurations: First, the **Baseline System** which consists of a standard multi-agent implementation without intrinsic memory. It uses standard prompt templates for each agent role, relying exclusively on conversation history for context. Second is our **Intrinsic Memory System** approach with agent-specific memories. It implements agent-specific memories and updates them intrinsically based on agent outputs, and constructs context using both conversation history and agent memories.

Both systems used identical agent roles and task specifications, with Llama-3.2-3b as the underlying LLM. Each agent role was initialized with the same role description and initial instructions across the two system configurations to ensure a fair comparison.

An agent selection function iterates through each worker agent and the conversation delegation agent (CDA), ensuring that all agents are represented in the discussion. Once all agents have accepted a

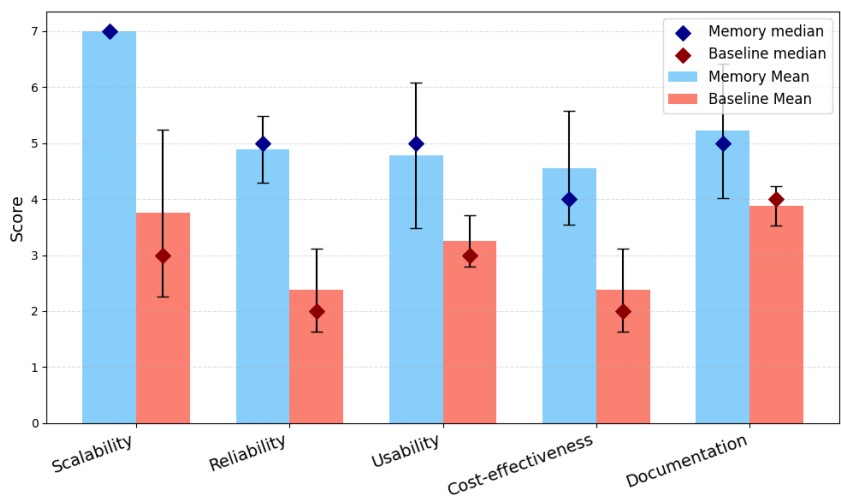

Figure 3: LLM-as-a-Judge metrics for the Data Pipeline design case study.

proposed solution, marked through the "ACCEPT" flag, the CDA emits a "FINALIZE" flag, prompting the Documentation Engineer Agent to produce the final data pipeline output. The full algorithm for finalisation and ordering of agents is displayed in the Appendix A Algorithm 2.

## 5.2 EVALUATION METRICS

To evaluate the quality of the data pipeline designs generated by our memory agents, and to compare to the pipeline designs generated from default Autogen, we use an LLM as a judge (Zheng et al., 2023) to score each pipeline design and provide a qualitative analysis to support these scores. We evaluated the multi-agent system performance under the following metrics:

- Scalability: ability for the data pipeline to handle increasing data volumes or user loads.
- Reliability: ability for the data pipeline to handle failures and ensure data integrity.
- Usability: is there enough detail in the data pipeline design for developers to implement the design?
- Cost-effectiveness: balance between costs and benefits of each chosen component.
- Documentation: how well-justified and documented is the choice of elements for the data pipeline?

The scalability and reliability metrics are chosen as core requirements in modern data pipelines. Scalability reflects the ability to grow the pipeline to handle larger volumes of data and users, while reliability ensures the pipeline is consistent and fault-tolerant, both of which are crucial if the pipeline were to be deployed. Usability and documentation metrics reflect the details and design decisions taken. A strong design is not useful if it does not contain enough detail or is too abstract to be practically implemented. Usability measures whether the output designs are detailed and clear enough for engineering teams to implement. Design decisions must be well-documented, with clear justifications and explanations for each component, which reveals the reasoning behind the agents' choices. Finally, the cost-effectiveness metric evaluates whether the pipeline design has considered and balanced the need for computation resources with the cost of those resources. Run-time metrics such as latency and throughput are not included in our evaluation metrics as we only present the design of the data pipelines to be evaluated, and do not implement the designs into code

## 5.3 DATA PIPELINE DESIGN PERFORMANCE

The median and standard deviation of each quality metric is presented in Figure 3. The Intrinsic Memory system shows consistent improvement on all metrics compared to the baseline Autogen.

Table 1: Mean efficiency and LLM-as-a-Judge metrics after 10 independent runs, with p-values calculated using a Wilcoxon ranked sum test. Usability and number of conversation turns is highlighted in italics as the metrics that do not show statistical significance between the baseline and our Intrinsic Memory approach.

| Metric | Baseline Autogen | Intrinsic Memory | p-value |
|---|---|---|---|
| Tokens | 36077 | 47830 | 0.0195 |
| *Conversation turns* | 14.3 | 16 | *0.2632* |
| Scalability | 3.75 | 7 | 0.0004 |
| Reliability | 2.37 | 4.9 | 0.0003 |
| *Usability* | 3.25 | 4.9 | 0.0093 |
| Cost-effectiveness | 2.37 | 4.7 | 0.001 |
| Documentation | 3.87 | 5.4 | 0.0077 |

The Documentation quality focuses on the clarity and how well-justified the design choices are. While Intrinsic Memory helps to boost the Documentation score over the baseline, the score is still relatively low at a mean of 4.9. This suggests that retaining memory of the conversation alone does not guarantee good justification, and while some context and attributes of each component are remembered, the reasons for choosing the components are not. This could be a problem with the training corpus, and a requirement for better annotated training data. Similarly, the Usability score is low with means of 3.32 and 4.9 for the baseline and Intrinsic Memory, respectively.

The improved quality comes at a cost of additional tokens outlined in Table 1. Intrinsic Memory uses on average 32% more tokens than the baseline as it outputs are more descriptive on average, although the number of conversation turns is similar and not statistically significant. This indicates that the addition of a memory module costs additional token overhead to maintain, but does not increase the number of conversation turns between agents.

## 5.4 QUALITATIVE ANALYSIS OF DATA PIPELINE OUTPUTS

Figure 4 shows snippets for one component from the highest-scoring outputs for the intrinsic memory agent system and baseline Autogen system.

The Intrinsic Memory Agent system outperforms the baseline system across the five quality metrics. In terms of scalability, the Intrinsic Memory Agent system is capable of providing an overall assessment of scalability, specifically around varying data volumes, whereas the baseline system encapsulates that measure only in the form of "maintenance difficulty" for each component of the pipeline. In terms of reliability, the Intrinsic Memory Agent provides considerations for each component, such as AWS Kinesis's secure streaming capabilities and considerations as well as the use of Docker containers within amazon SageMaker to improve stability and reproducibility of ML pipelines. The Intrinsic Memory Agent provides a more descriptive Usability output of the Intrinsic and a clearer pathway to implementation. In terms of cost, the Intrinsic Memory Agent makes specific calculations and observations for the cost-effectiveness and resource requirements, including reasoning behind each component choice, whereas the baseline system limits itself to overall evaluations of implementation and maintainability difficulties.

Finally, the Intrinsic Memory Agent ultimately provides justification and documents its recommendation under each component, including pros and cons for each component choice.

Overall, the Intrinsic Memory Agent provides a more descriptive answer and more value to engineers by specifying tools, configurations and trade-offs. For example, its Data Streaming design recommends Amazon Kinesis, whereas the baseline simply states "Ingest data from various sources (camera, lidar, radar) at high speeds." Similarly, the IMA cites the specific connections between components that must be implemented (for example, Amazon S3 =¿ Amazon EC2 through API) . Although some precise configuration settings remain unspecified, the baseline merely names each component without offering implementation details or alternatives.

The components specified by the Intrinsic Memory Agent are more relevant to the problem specification. The data pipeline design task explicitly specifies the input data contains lidar and radar data

```
"Component 1":  "Data Ingestion (Amazon S3)"
"AWSName":  "AmazonS3",
"Pros":  ["Scalable", "durable", "secure storage for raw data"],
"Cons":  ["Additional cost for storing large amounts of data"],
"Design":  "Use S3 as a central repository for all data sources,
with separate buckets for each source if needed.",
"Details":  "Implement S3 event notifications to trigger
processing workflows upon new data arrival."
```

(a) Intrinsic Memory Agent system sample from highest-scoring output. This data pipeline received scores of Scalability: 7, Reliability: 5, Usability: 6, Cost-effectiveness: 6, Documentation: 7

```
"Component 1":
 "Name":  "Data Ingestion",
 "Description":  "Ingest data from various sources (camera,
lidar, radar) at high speeds",
 "Implementation difficulties":  7,
 "Maintainability difficulties":  6
```

(b) Baseline Autogen system system sample from highest-scoring output. This data pipeline received scores of Scalability: 5, Reliability: 4, Usability: 3, Cost-effectiveness: 3, Documentation: 2

Figure 4: Snippets of one component within the data pipeline design from both systems. The full outputs can be found in the appendix in Figures 11 and 13.

sources, in which SageMaker and Kinesis are particularly relevant, specifically used for lidar data processing. This contrasts the vague "Lidar data processing" component of the baseline, which only contains a general description to process the data without providing any details.

# 6 DISCUSSION AND LIMITATIONS

Although the Intrinsic Memory Agent approach shows improved performance across the data pipeline generation task and the selected benchmarks, further validation is required across a broader set of complex tasks, potentially with varying number of agents, and models. Furthermore, our approach's performance and consistency comes at the cost of increased token usage, due to the additional update calls. Further work to reduce the number of update calls, updating only when necessary, will help to alleviate the additional usage.

The results demonstrate that a movement towards heterogeneity of agents leads to an improvement in performance of the multi-agent system, allowing agents to focus more specifically on an area of the design. This indicates that methods to provide additional heterogeneity, such as the ability to fine-tune agents towards their specialisation, might see additional performance gains, alongside the personalization of memories focused on individual experience.

# 7 CONCLUSION

This paper introduces Intrinsic Memory Agents, a novel multi-agent LLM framework that constructs agent-specific heterogeneous memories to enhance multi-agent collaboration in discussion and planning tasks. Evaluation PDDL dataset and on a practical data pipeline design problem demonstrates our framework's improved performance on structured planning tasks, with a 15.5% increase over the next best memory architecture, at the cost of increased token usage. Further evaluation on the ALF-World and FEVER datasets demonstrates our approach's consistent performance, ranking among the top memory mechanisms with the lowest standard deviation while the best memory mechanisms on these benchmarks show high standard deviation. Our strong performance using a generic memory template demonstrates the generalisablity of our approach to other problems, without the need to hand-craft high quality memory templates.

Results on the data pipeline case study further show the Intrinsic Memory Agents' enhanced ability to collaborate on complex tasks. The Intrinsic Memory Agent system outperforms the baseline system across all quality measures of scalability, reliability, usability, cost-effectiveness, and documentation, as well as an ability to more closely follow the task specification, providing more actionable recommendations by suggesting specific tools and frameworks, as well as trade-off details of each component in the pipeline.

REPRODUCIBILITY STATEMENT

We have taken care to ensure that our experiments and results are transparent and reproducible by detailing the models, computational setup, code, statistical tests, and prompts used in our experiments. The LLM model (Llama3.2:3b) is cited, named, and referenced in the main text. The computational infrastructure used, including the GPU model names and operating system are specified in section 4 of the main text. For code, the names and versions of relevant Python libraries are specified within the supplementary code files. A Wilcoxon rank-sum test is used to test statistical significance for the data pipeline case study. P-values and standard deviation measures are included in the performance analysis in section 5.3. 5 independent runs with different set seed are used for the numeric benchmarks, with the seeds specified within the supplementary code. All code for running the data pipeline case study and numeric benchmarks are included in the supplementary materials. Finally, the selected prompts of the multi-agent and Intrinsic memory architecture are shown in Appendix D. Further prompts for each agent can be found as part of the supplementary code, under the "prompts" directory.

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

# A  ALGORITHMS

This section contains the context construction algorithm presented in Section 3 and the finalisation algorithm presented in 5.1.

---

**Algorithm 1** The context construction algorithm, which takes the current conversation history, memory of the agent, and maximum number of tokens. It appends the most recent conversation turn and agent memory to the context first, before using the remainder of the tokens to append the rest of the conversation history, ensuring the memory and most recent output is always included.

---

```python
def construct_context(conversation_history,
                      agent_memory,
                      max_tokens):
context = []

# Include the initial task description
context.append(conversation_history[0])
context.append(agent_memory)

# Add most recent conversation turns until context limit is reached
remaining_tokens = max_tokens - count_tokens(context)
recent_turns = []

for turn in reversed(conversation_history[1:]):
    turn_tokens = count_tokens(turn)
    if turn_tokens <= remaining_tokens:
        recent_turns.insert(0, turn)
        remaining_tokens -= turn_tokens
    else:
        break

context.extend(recent_turns)
return context
```

---

**Algorithm 2** The finalisation algorithm that specifies the order of agents speaking. It is a modified round-robin discussion between the agents: The discussion begins with each of the worker agents (BOA, DEA, MLA, IA) contributing to the conversation, with each worker's turn being followed by the conversation delegation agent (CDA). Once the workers have each had their turn, the knowledge integration agent and evaluation agent make their contributions, and the cycle begins again. The CDA is programmed to dedicate a certain number of turns to discussion, proposals, and consensus. The number of turns dedicated to each conversation stage is tracked, and once the consensus round is reached, each agent is asked to confirm if they agree with the proposed solution or not. If all agree on the proposed solution as being acceptable, the CDA will emit a "FINALIZE" response, triggering the documentation joining agent (DJE) to compile the agreed response and format it according to the task requirements.

```python
workers = [BOA, DEA, MLA, IA]
global turn_counter
turn_counter += 1

if "FINALIZATION" in groupchat.messages[-1]['content']:
    return DJE
if last_speaker is CDA:
    global worker_counter
    w = workers[worker_counter%4]
    worker_counter += 1
    print(f'worker_counter:■{worker_counter}')
    return w
elif last_speaker in workers:
    if worker_counter %4 == 0:
        return KIA
    else:
        return CDA
elif last_speaker is KIA:
        return ERA
elif last_speaker is ERA:
        return CDA
```

# B   ABLATION STUDY

We conduct an ablation study to understand the sensitivity of the Intrinsic Memory Agents to the structure of the templates that the agents use to update their memory. We evaluate three approaches:

1. **Manual template** - Manual hand-crafted templates for each type of dataset and problem, based on the kind of task the agents have to solve.

2. **Generic template** - A generic template to store updated memory without any reference to specific problems or fields.

3. **LLM-generated template** - We prompt the agent LLM to self-generate a template based on the instructions of the task, allowing the agent to dynamically create templates it deems suitable for each task.

The approaches are evaluated on the PDDL, FEVER, and ALFWorld benchmark problems, tested across different base language models with multiple independent runs using set seeds for reproducibility and consistency.

## B.1   TEMPLATE APPROACHES

### B.1.1   MANUAL TEMPLATE

In the manual template approach, a strict structured template is used that agents must follow to update their internal memory. The manual templates are created to track the relevant information of the given task, with specific fields.

### B.1.2   GENERIC TEMPLATE

For the generic approach, we use a universal template structure, giving only general fields to be filled in with the current task description and trajectory, allowing the model to use any form of memory representation, and not strictly requiring the same structured format to be used each time.

### B.1.3   LLM-GENERATED TEMPLATE

Finally, we leverage each agent's own capabilities to create a template during initialisation which it deems suitable for the task at hand. This approach removes the need to create manual templates for each type of task while allowing for memory templates specific to the task. The template generation prompt is as follows:

## B.2   ABLATION RESULTS

We perform the ablation study on two additional models: Gemma3-12b and Mistral-7b. Tables 2 and 3 display the mean and standard deviation of each template approach on the PDDL, FEVER, and ALFWorld benchmarks. We find that the approaches using an LLM-generated template or a generic template outperform the manual template approach. In general, the generic template performs best, but performance for both the LLM-generated approach and generic approach are similar using Gemma3. Token usage varies based on the underlying model. For example, Mistral uses more tokens on average for the generic approach whereas Gemma uses more tokens for the LLM-based approach. Note that the cost of the template generation in the LLM approach is a fixed one-time cost at the beginning of each problem, and therefore adds little overhead compared to the rest of the runtime.

```
You are a MEMORY UPDATER for a PDDL-style planning agent.
Your job: – Maintain a compact JSON memory capturing stable,
reusable information across tasks and domains. – Only store
information that improves future planning: common strategies,
mistakes, valid action patterns, state-transition insights.
– Do not store long histories. Keep everything concise and
deduplicated.
Inputs you receive each update call: – currentmemory: the
previous memory as json (may be empty re-init). – latestturn: the
agent's most recent Thought/Action/Observation. – currenttask:
one of blockworld, barman, gripper, tyreworld. – goal: current
goal description.
OUTPUT: – Return ONLY the updated memory as valid JSON following
the template below. – No extra commentary.
------------------------ MEMORY TEMPLATE (ALWAYS FOLLOW)
"task summary": "brief description of PDDL planning setting",
"global strategies": [ "high-level reusable planning heuristics
across domains" ], "domains":    "blockworld":    "valid action
patterns": ["pickup X", "putdown X", "stack X Y", "unstack X Y"],
"good strategies": ["free target block before stacking"], "invalid
patterns": ["wrong think format", "stack without clear base"],
"mistakes": ["attempting pickup while arm full"] , "barman":
"valid action patterns": ["hand grasp glass", "fill-shot ...",
"pour-shot-to-clean-shaker ..."], "good strategies": ["ensure
hand availability before filling"], "invalid patterns": ["fill
without holding glass"], "mistakes": ["grasp with occupied hand"]
, "gripper":    "valid action patterns": ["move R1 R2", "pick O
Room Gripper", "drop O Room Gripper"], "good strategies": ["carry
multiple items before moving rooms"], "invalid patterns": ["drop
object in wrong room"], "mistakes": ["pick while gripper full"]
, "tyreworld":    "valid action patterns": ["open X", "fetch O
C", "loosen N H", "jack-up H"], "good strategies": ["open boot
early to access tools"], "invalid patterns": ["loosen nut without
wrench"], "mistakes": ["inflate wheel without pump"]  , "tasks":
[ "id": "identifier or hash of goal", "goal": "exact goal text",
"status": "pending|solved", "helpful observations": ["short state
insights from valid steps"], "invalid actions": ["summaries of
failed attempts"], "progress notes": ["short planning insights for
this task"]  ]
------------------------ UPDATE INSTRUCTIONS
1. Parse current memory. – If empty or invalid, initialize using
the template above.
2. Update the domain-specific sections: – From latest turn, add
new useful action patterns, invalid patterns, or mistakes. – Keep
lists short, deduplicated, and generalisable.
3. Update global strategies if the latest turn reveals a robust
cross-domain heuristic.
4. Update the relevant task entry: – If no entry exists for this
goal, create one. – Add helpful observations if new actionable
state insights appear. – Add invalid actions if latest turn
shows an invalid move. – Add progress notes for general reasoning
improvements. – If task finished, mark status = "solved".
5. Return ONLY the updated JSON memory, nothing else.
```

Figure 5: Manually generated prompt for the LLM agent in the PDDL task.

```
Use your latest response to populate and update the current memory
with factual information to solve the task based on the task
description.

## Task Description:

{task_description}

## Current Task Trajectory:

{task_trajectory}

## Current Memory:

{current_memory}
```

Figure 6: Prompt for the LLM agent to update its memory

```
I have an AI agent that has to complete a task.
The agent has a memory that is updated each time the LLM responds
by comparing the latest response and the existing memory, and
adding any new important information.  The memory should be
templated based on the nature of the task following a json-style
format.  The memory update is conducted as a prompted LLM call to
update the memory.  Provide the instructions to the agent for such
an update operation, as well as the generic memory template for
this particular task.  Provide the full answer as a single prompt.
Only include the most crucial details to the updating instructions
to preserve token usage.  Do not explain or describe the prompt,
simply return the prompt and nothing more.
This is the task description:  {task_description}
```

Figure 7: Prompt for the LLM agent to generate its own memory template.

| Benchmark | Memory | Rewards | | Average tokens |
|-----------|--------|---------|-----|----------------|
| | | Mean | Std | |
| | Intrinsic | 0.063648 | 0.004113 | 848,052 |
| PDDL | Intrinsic-LLM | **0.069740** | 0.010224 | 609,102 |
| | Intrinsic-Generic | 0.066198 | 0.008096 | 613,631 |
| | Intrinsic | 0.119974 | 0.024228 | 1,395,395 |
| FEVER | Intrinsic-LLM | 0.300746 | 0.051672 | 949,377 |
| | Intrinsic-Generic | **0.379853** | 0.041365 | 1,079,041 |
| | Intrinsic | 0.015390 | 0.000317 | 1,789,989 |
| ALFWorld | Intrinsic-LLM | 0.014944 | 0.000046 | 991,884 |
| | Intrinsic-Generic | **0.029926** | 0.000116 | 1,027,602 |

Table 2: Mistral:7b

| Benchmark | Memory | Rewards | | Average tokens |
|---|---|---|---|---|
| | | Mean | Std | |
| PDDL | Intrinsic | 0.253164 | 0.049818 | 379,786 |
| | Intrinsic-LLM | 0.253986 | 0.032846 | 359,544 |
| | Intrinsic-Generic | **0.260255** | 0.022382 | 350,541 |
| FEVER | Intrinsic | 0.608603 | 0.006874 | 178,456 |
| | Intrinsic-LLM | 0.649391 | 0.004576 | 86,849 |
| | Intrinsic-Generic | **0.653500** | 0.005910 | 88,274 |
| ALFWorld | Intrinsic | 0.025065 | 0.000262 | 942,916 |
| | Intrinsic-LLM | 0.045002 | 0.000350 | 883,913 |
| | Intrinsic-Generic | **0.048296** | 0.008294 | 879,134 |

Table 3: Gemma3:12b

## C    FULL BENCHMARKING RESULTS

| Benchmark | Memory | Rewards | | Average tokens |
|---|---|---|---|---|
| | | Mean | Std | |
| ALFWorld | ChatDev | 0.020904 | 0.006491 | 174,451 |
| | Empty | 0.020408 | 0.000000 | 113,817 |
| | G-Memory | 0.040851 | 0.019529 | 104,492 |
| | Generative | 0.060656 | 0.031094 | 181,211 |
| | MemoryBank | 0.031623 | 0.004932 | 160,273 |
| | MetaGPT | 0.030777 | 0.000547 | 136,716 |
| | Voyager | **0.072112** | **0.034554** | 102,482 |
| | *Intrinsic-Generic* | 0.048296 | 0.008294 | 784,119 |
| | *Intrinsic-LLM* | 0.045002 | 0.000350 | 882,158 |
| FEVER | ChatDev | 0.617500 | 0.016583 | 80,656 |
| | Empty | 0.646667 | 0.015706 | 55,539 |
| | G-Memory | 0.628792 | 0.027763 | 267,823 |
| | Generative | 0.651250 | 0.011087 | 117,278 |
| | MemoryBank | 0.632839 | 0.010771 | 65,997 |
| | MetaGPT | **0.667000** | **0.028853** | 48,998 |
| | Voyager | 0.643000 | 0.025884 | 62,555 |
| | *Intrinsic-Generic* | 0.653500 | 0.005910 | 88,274 |
| | *Intrinsic-LLM* | 0.649391 | 0.004576 | 86,849 |
| PDDL | ChatDev | 0.222746 | 0.021114 | 70,765 |
| | Empty | 0.224329 | 0.019766 | 52,075 |
| | G-Memory | 0.152222 | 0.039180 | 162,712 |
| | Generative | 0.164944 | 0.021977 | 84,113 |
| | MemoryBank | 0.158083 | 0.013697 | 66,299 |
| | MetaGPT | 0.197106 | 0.018546 | 92,436 |
| | Voyager | 0.191088 | 0.031562 | 68,483 |
| | *Intrinsic-Generic* | **0.260255** | **0.022382** | 352,301 |
| | *Intrinsic-LLM* | 0.253986 | 0.032846 | 359,302 |

Table 4: Rewards and tokens for each memory framework on the three benchmark problem sets.

## D  FULL PROMPTS AND EXAMPLE OUTPUTS

```
You are maintaining the memory of an agent working as [ROLE]
in a multi-agent conversation.  Use the old memory and the
newest output by the agent to populate and up- date the
current memory json with factual information.

For context, old memory content:
[MEMORY_CONTENT]

Current content generated by the agent:
[AGENT_OUTPUT]

Update the memory content to incorporate new information
while preserving key historical context.  The updated content
should be concise and focus on information relevant to both
the old memory and the newly generated output.
```

Figure 8: Prompt of the memory update function, where *ROLE* is the agent's role specification $R_n$; *MEMORY_CONTENT* is the current content $M_{n,m-1}$; *AGENT_OUTPUT* is the agent's output $O_{n,m}$.

```
''' This discussion session is set up to discuss the best data
pipeline for a real time data intensive machine learning training
and inference self driving application.  The goal is to discuss and
find consensus on how to set up the data pipeline, including each
component in the data pipeline.
You can assume that we have access to AWS.

**Data Description:** Real-time data of cars driving in street.
There are 6 camera sources with data in .jpg format; 1 lidar source
in .pcd.bin format; and 5 radar sources with data in .pcd format.

**Discussion and Design:**
- Emphasize comprehensive understanding of the data sources,
processing requirements, and desired outcomes.
- Encourage each other to engage in an open discussion on potential
technologies, components, and architectures that can handle the
diverse data streams and real-time nature of the data.
- Keep the conversation on design and evaluating the pros and
cons of different design choices, considering scalability,
maintainability, and cost-effectiveness.
- The team should agrees on a final architectural design,
justifying the choices made.
- The team should produce the required the document
PIPELINE_OVERVIEW.json.

**Final Output:**
- Produce a concise summary of the agreed-upon pipeline
architecture, highlighting its key components and connections.
- Provide a high-level plan and rationale for the design,
explaining why it is well-suited for the given data and use case.
- Estimate the cloud resources, implementation efforts, and
associated costs, providing a rough breakdown and complexity
rating.
- Generate a `PIPELINE_OVERVIEW.json` file, detailing the proposed
complete architecture in JSON format with the following fields:
- \Platform\:  A cloud service provider's name if the cloud
solution is the best, or \local server" if locally hosted servers
are preferred.
- \Component 1":  The first component in the pipeline framework.
- \Component 2":  The second component in the pipeline framework.
Continue until all required components are listed.
- \Implementation difficulties":  A rating from 1 to 10 (lowest to
highest).
- \Maintenance difficulties":  A rating from 1 to 10 (lowest to
highest).

**Instructions:**
- Remember, this is a collaborative design discussion, not a
project execution.  Refrain from assigning tasks with deadlines.
- Keep the conversation focused on architectural choices,
technologies, and potential challenges.
- Emphasize the importance of a well-thought-out design.  '''
```

Figure 9: Prompt for the Data Pipeline design. This is the task description provided to all agents in the system.

```
```
You are an expert in data pipeline design evaluation.  Your
task is to evaluate the following data pipeline design based
on its description.  For each of the specified metrics,
assign a score from 1 to 10, where 1 is the lowest and 10
is the highest.  Provide a brief justification for each
score.  Be critical and harsh if the design is poor and give
it a low score.  Base your evaluation solely on the provided
description.  Do not assume any additional information.

Metrics:
1.  Scalability:  Ability to handle increasing data volumes
or user loads.
2.  Reliability:  Ability to handle failures and ensure data
integrity.
3.  Usability:  Enough detail for developers to implement the
design.
4.  Cost-effectiveness:  Balance between costs and benefits.
5.  Documentation:  How well-justified is the choice of
elements for the data pipeline

Provide your evaluation in the following format in a json
dict:
[metric1]:  score:  [score],
justification:  [justification],
[metric2]...
```
```

Figure 10: Prompt for the LLM-as-a-Judge to evaluate data pipeline designs.

```
"Platform":  "AWS"

"Component 1":  "Amazon S3 (Data Storage)"
"pros":  "Scalable, durable, and secure storage for raw data."
"cons":  "Additional cost for storing large amounts of data."
"design":  "Use S3 as a central repository for all data sources,
with separate buckets for each source if needed."
"details":  "Implement S3 event notifications to trigger data
processing workflows upon new data arrival."

"Component 2":  "Amazon EC2 (GPU Instances) | Image & Point Cloud
Processing"
"pros":  "Scalable, high-performance computing resources for
complex image and point cloud processing."
"cons":  "Higher cost due to instance usage and GPU pricing."
"design":  "Use EC2 NVIDIA GPU instances for camera source data
processing."
"details":  "Implement Docker-based containerization for
reproducibility and efficient execution."

"Component 3":  "Amazon SageMaker | Model Training & Processing"
"pros":  "Fully managed, high-performance infrastructure for ML
training and inference."
"cons":  "Higher cost for GPU-backed instances."
"design":  "Use SageMaker for lidar and radar data processing,
including preprocessing and feature extraction."
"details":  "Use Docker containers to ensure stable, reproducible
ML pipelines."

"Component 4":  "Amazon Kinesis | Real-Time Streaming"
"pros":  "Scalable, fast, and secure streaming and analytical
capabilities."
"cons":  "Additional cost for high-throughput streaming."
"design":  "Use Kinesis for real-time ingestion, analytics, and
event-driven triggers."
"details":  "Implement aggregation, filtering, and visualization
workflows to support decisions."

"Component 5":  "Amazon Redshift | Batch Analytics"
"pros":  "High-performance analytics engine with strong scaling
capabilities."
"cons":  "Additional cost for warehousing large datasets."
"design":  "Use Redshift for batch analytical workloads across
multi-source data."
"details":  "Implement analytics queries and visualization
pipelines for reporting and BI."
```

Figure 11: Example output from intrinsic memory agent system which scored the highest. The intrinsic memory agent system adheres to the initial prompt well, providing an answer in the correct format, as well as providing sufficient justification for the choice of components, as well as implementation details.

```
"Component 6":  "Amazon IAM | Identity & Access Control"
"pros":  "Secure, scalable identity and permissions management."
"cons":  "Requires careful design to avoid security
misconfiguration."
"design":  "Implement IAM roles and policies to manage access
across all pipeline components."
"details":  "Use Cognito for user authentication and authorization
where appropriate."

"Component 7":  "Amazon Cognito | Authentication"
"pros":  "Managed, secure identity federation and authentication."
"cons":  "Additional cost depending on usage tiers."
"design":  "Use Cognito for managing user identity, login, and
token issuance."
"details":  ""
"Connections 1":  "Amazon S3 → Amazon EC2 (GPU instances) (API)"
"Connections 2":  "Amazon EC2 (GPU instances) → Amazon SageMaker
(API)"
"Connections 3":  "Amazon SageMaker → Amazon Kinesis (API)"
"Connections 4":  "Amazon Kinesis → Amazon Redshift (API)"
"Connections 5":  "Amazon IAM → Amazon Cognito (API)"

"Potential Challenge 1":  "Scalability"
"Potential Challenge 2":  "Performance"
"Potential Challenge 3":  "Security"

"Rationale":  "Pipeline is designed to be scalable, maintainable,
and secure using microservices-based architecture."

"Complexity Rating":  "7"

"Estimated Cloud Resource 1":  "S3:  10{20 TB of storage"
"Estimated Cloud Resource 2":  "EC2 (GPU instances):  4{8
GPU-enabled instances"
"Estimated Cloud Resource 3":  "SageMaker:  2{4 GPU-enabled
instances"
"Estimated Cloud Resource 4":  "Kinesis:  2{4 streams"
"Estimated Cloud Resource 5":  "Redshift:  2{4 clusters"

"Implementation Effort":  "12{16 weeks"

"Cost 1":  "S3:  $500{$2,000 per month"
"Cost 2":  "EC2 (GPU instances):  $5,000{$20,000 per month"
"Cost 3":  "SageMaker:  $2,000{$10,000 per month"
"Cost 4":  "Kinesis:  $1,000{$5,000 per month"
"Cost 5":  "Redshift:  $2,000{$10,000 per month"
```

Figure 12: Example output from intrinsic memory agent system which scored the highest. The intrinsic memory agent system adheres to the initial prompt well, providing an answer in the correct format, as well as providing sufficient justification for the choice of components, as well as implementation details. (Continued)

```
  "Platform":  "AWS",
  "Component 1":
  "Name":  "Data Ingestion",
  "Description":  "Ingest data from various sources (camera, lidar,
  radar) at high speeds",
  "Implementation difficulties":  7,
  "Maintainability difficulties":  6 ,
  "Component 2":
  "Name":  "Image Processing",
  "Description":  "Process camera data in real-time to extract
  relevant features",
  "Implementation difficulties":  8,
  "Maintainability difficulties":  7

  ,
  "Component 3":
  "Name":  "Object Detection",
  "Description":  "Detect objects of interest (e.g., pedestrians,
  cars, lanes) within image frames and track their movement over
  time",
  "Implementation difficulties":  9,
  "Maintainability difficulties":  8

  ,
  "Component 4":
  "Name":  "Lidar Data Processing",
  "Description":  "Process lidar data in real-time to estimate
  distances, velocities, and other relevant metrics",
  "Implementation difficulties":  8,
  "Maintainability difficulties":  7

  ,
  "Component 5":
  "Name":  "Radar Data Processing",
  "Description":  "Process radar data in real-time to estimate
  distances, velocities, and other relevant metrics",
  "Implementation difficulties":  8,
  "Maintainability difficulties":  7

  ,
  "Component 6":
  "Name":  "Data Fusion",
  "Description":  "Fuse the outputs from different sensors (camera,
  lidar, radar) to create a more accurate representation of the
  environment",
  "Implementation difficulties":  9,
  "Maintainability difficulties":  8

  ,
  "Component 7":
  "Name":  "Model Training",
  "Description":  "Train machine learning models on large datasets
  using AWS SageMaker's Training Grounds feature",
  "Implementation difficulties":  8,
  "Maintainability difficulties":  7 ,
  "Component 8":
  "Name":  "Inference",
  "Description":  "Perform real-time inference on trained models,
  making predictions on new, unseen data",
  "Implementation difficulties":  9,
  "Maintainability difficulties":  8
```

Figure 13: Example output from baseline Autogen system which scored the highest.