# OpenReview forum: "Intrinsic Memory Agents: Heterogeneous Multi-Agent LLM Systems through Structured Contextual Memory"
_ICLR.cc/2026/Conference — Submitted to ICLR 2026_

### Official Review · Reviewer_iVU5 · 2025-10-15

**Soundness:** 3
**Presentation:** 3
**Contribution:** 2
**Rating:** 2
**Confidence:** 2

**Summary:**

The paper proposes a new system to augment LLMs with contextual memory.

There are lots of systems that attempt to do something similar, the main contribution here is the use of a "multi-agent system", where there are several agents responsible for storing the structured memories relevant for different contexts.

The idea is overall reasonable, though the paper is fairly non-techhnical, and the baselines fairly limited.

**Strengths:**

Addresses an overall important problem and limitation in existing LLMs

Specific and reasonable solution and methodological contribution, difference compared to existing approaches is quite clear

**Weaknesses:**

Paper is highly non-technical, it presents an idea but it's a fairly simple system that mightn't meet the bar as a technical contribution for ICLR

Most critically, the technical evaluation is *very* limited. Just a single baseline (it's hard to figure out exactly what this is?), and a data pipeline that seems to be of the author's own invention. Not much by way of comparison to what seems like a "state-of-the-art" memory system. This seems mostly a dealbreaker for acceptance to ICLR

**Questions:**

Can you provide more detail about the specifics of the baseline? Why can't other baselines be considered? Can the method be evaluated on standard datasets that are widely used? Can it be compared to other structured memory systems?

---

> ### Author Response · Authors · 2025-11-21
>
> Thank you for your comments. Below we provide further elaboration on the contribution our paper makes to memory management in multi-agent systems.
>
> ### Technical justification
> We acknowledge that our approach is based at the prompt level, using the agent’s own reasoning to perform memory update steps. Our approach solves issues of role adherence and instruction following in multi-agent systems using an approach that can be widely applied to existing multi-agent systems. We will further provide examples of the mechanism’s impact in the data pipeline case study, to show where and how our mechanism improves performance compared to a baseline multi-agent system. Our quantitative benchmark performance further shows the generalisability of our method to different domain tasks. We improve on our existing benchmarking by evaluating an ablation on the template structure, across multiple independent runs, multiple underlying models, and two more benchmark datasets (FEVER and ALFWorld).
>
> ### Baseline specifics
> We would be grateful if you could clarify what was meant by “baseline”. We are not sure whether baseline refers to the choice of a data pipeline planning task, the choice of the PDDL benchmark, or the choice of baseline Autogen as our comparison for that task.
>
> ### Dataset evaluation
> We have compared against other structured memory systems in our benchmarking, with 6 different memory systems. As part of our additional benchmarking and ablation, we will add the FEVER and ALFWorld datasets to the benchmarking, for more standard multi-agent benchmarking and to show the generalisability of our method. We will do this over multiple runs, reporting the mean and standard deviation for additional confidence in our results.

---

### Official Review · Reviewer_uNdj · 2025-10-30

**Soundness:** 2
**Presentation:** 2
**Contribution:** 2
**Rating:** 4
**Confidence:** 4

**Summary:**

This paper proposes the Intrinsic Memory Agents framework for MAS. IMA tackles the memory decay and role drift, which are caused by constrained context windows. It achieves this by introducing structured, agent-specific memory templates that dynamically update based on the agent's outputs, ensuring memory remains concise and role-aligned for complex tasks.

**Strengths:**

The proposed output-driven memory evolution offers a specialized solution to the context window problem in MAS. This mechanism helps maintain role adherence by constantly refreshing agent-specific context, which benefits the performance of MAS on two downstream tasks.

**Weaknesses:**

1. The core mechanism of IMA—refining a structured memory template based on self-output, primarily operates at the prompt engineering level. The design lacks a distinct technical contribution, relying entirely on an LLM's internal reasoning step. The authors must rigorously justify why this intrinsic, prompt-based refinement constitutes a novel advance beyond existing context management techniques.

2. The experimental validation is limited to a specific set of collaborative tasks, which restricts the assessment of IMA's utility for general MAS problems. To credibly claim the framework's utility for complex collaborative problem-solving, evaluation should extend to more representative MAS benchmarks like coding.

3. The authors are encouraged to provide explicit, documented examples showing some bad cases of baseline agent memory updating methods, and how IMA's structured memory successfully prevents that specific failure mode. This is essential for understanding the mechanism's real-world impact.

4. The experimental validation is confined to a standard, sequential MAS system. The claimed benefits of agent-specific memory management need rigorous discussion regarding its universality across more complex MAS backbones.

**Questions:**

1. Is the performance of IMA  dependent on the capability of the underlying LLM? Please provide an ablation study showing the performance of the full framework when substituting the primary LLM.

2. Has the author studied the influence of agent numbers on the effectiveness of the proposed method?

---

> ### Author Response · Authors · 2025-11-21
>
> Thank you for your comments, in our ablation study, we include additional benchmarking of the FEVER and ALFWorld datasets to show generalisability of our method. The FEVER dataset tests of fact extraction and verification, while the ALFWorld dataset is a benchmark for learning a mapping from natural language instructions to sequences of actions.
>
> ### Novel justification
> We acknowledge that our approach is based at the prompt level, using the agent’s reasoning to perform memory update steps. We argue that our approach is novel as it solves issues of role adherence and instruction following in multi-agent systems using an approach that can be widely applied to existing multi-agent systems.
> We will further provide documented examples of the mechanism’s impact in the data pipeline case study, to show where the baseline context of agents fails and how our mechanism improves performance in these cases. We backup our claims through quantitative benchmarks, which we comprehensively expand upon, evaluating an ablation on the template structure, across multiple independent runs, multiple underlying models, and two more benchmark datasets (FEVER and ALFWorld).
>
> ### Ablation on substituting the underlying LLM
> Yes, as part of our update to run more benchmarks, with more independent runs, we will also include multiple underlying LLMs to provide an overview of the Intrinsic Memory system across different models. Further, we will run the data pipeline case study on a more powerful model such as GPT-4o to determine performance compared to the small language models. However, we note that the purpose and motivation of our Intrinsic Memory system is to address the limited context and performance of small language models, which becomes exacerbated in a multi-agent system. A larger powerful model has fewer limitations processing the large contexts generated by a multi-agent system conversation.
>
> ### Influence of agent numbers
> We acknowledge the number of agents within the system could have an effect on our proposed Intrinsic Memory. However, the full evaluation of this effect would require a comprehensive evaluation of multi-agentic architectures, agent numbers, and underlying models to ensure fair comparisons, which is out of scope for this work. We aim to improve the design of the memory mechanism and its comparison to other memory mechanisms rather than the architecture of the multi-agent systems. All baseline systems without memory and the system with our Intrinsic Memory use the same number of agents and multi-agent architecture for fair comparisons. Currently, the data pipeline case study uses 8 distinct agents while the quantitative benchmarks use 2, and our method's performance on both suggest a robustness to the number of agents in the system.

---

### Official Review · Reviewer_m3MB · 2025-10-31

**Soundness:** 3
**Presentation:** 3
**Contribution:** 2
**Rating:** 4
**Confidence:** 3

**Summary:**

The paper addresses context window limitations in multi-agent LLM systems , which impair memory consistency and role adherence. It introduces Intrinsic Memory Agents (IMA), a framework using structured, agent-specific memory templates. Unlike other methods, these memories are updated "intrinsically" from each agent's own output rather than via external summarization. This approach is designed to maintain heterogeneous, role-aligned perspectives. The method is evaluated on the PDDL benchmark, reportedly showing a 38.6% improvement over state-of-the-art memory approaches , and on a complex data pipeline design task, where it outperforms a baseline on 5 quality metrics.

**Strengths:**

1. Clear motivation and problem framing: The paper clearly articulates the problem it aims to solve and provides a strong rationale for its approach.

2. Clear and organized presentation: The paper is well-organized and uses language that is easy to understand, making the core concepts accessible.

3. Provides detailed experimental details for reproducibility: The inclusion of specific details, such as the prompts used (Appendix B), is commendable and supports the reproducibility of the work.

**Weaknesses:**

1. Limited experimental evaluation: The evaluation is confined to the PDDL benchmark and a single case study. In contrast, other significant works in this area (e.g., G-Memory) are often evaluated on a more diverse set of benchmarks spanning multiple domains.

2. Limited statistical robustness of the PDDL benchmark: The PDDL benchmark results are based on a single run with a set seed (Sec 4). This severely limits the statistical significance of the results presented in Table 1 and makes it difficult to assess the robustness of the performance gains.

3. Reliance on manual, task-specific template engineering: The framework's performance heavily depends on structured memory templates that are created manually for each specific task. The paper suggests "automated or generalized" methods as future work but provides no ablation study on the sensitivity to template design. It is unclear how much of the performance gain is due to the core intrinsic memory concept versus having a perfectly-tuned, hand-crafted template.

Reference

[1]Zhang, Guibin, et al. "G-Memory: Tracing Hierarchical Memory for Multi-Agent Systems." arXiv preprint arXiv:2506.07398 (2025).

**Questions:**

1. Could the evaluation be expanded to include more standard multi-agent benchmarks, such as ALFWORLD, HotpotQA, or FEVER, to demonstrate the method's generalizability?

2. Could the authors re-run the PDDL experiments (Sec 4) with multiple different random seeds for all compared methods? This would provide statistically robust results (e.g., mean and standard deviation) and add necessary confidence to the claims in Table 1.

3. Could the authors conduct an ablation study to analyze the framework's sensitivity to the quality of the manually created templates? For instance, how does performance degrade if a sub-optimal or more generic template is used? This would help isolate the true contribution of the intrinsic update mechanism.

Reference

[1]Shridhar, Mohit, et al. "Alfworld: Aligning text and embodied environments for interactive learning." arXiv preprint arXiv:2010.03768 (2020).

[2]Yang, Zhilin, et al. "HotpotQA: A dataset for diverse, explainable multi-hop question answering." arXiv preprint arXiv:1809.09600 (2018).

[3]Thorne, James, et al. "FEVER: a large-scale dataset for fact extraction and VERification." arXiv preprint arXiv:1803.05355 (2018).

---

> ### Author Response · Authors · 2025-11-21
>
> Thank you for your time and comments, we expand our additional benchmarking to include FEVER and ALFWorld, with some preliminary results shown in the overview response to demonstrate our approach's performance.
>
> ### More benchmarks
> Yes, we will conduct more runs on the FEVER and ALFWorld benchmarks to demonstrate the capabilities and generalizability of the system. If we have time we will include HotpotQA benchmarks, but given the size and length of the planned ablation study (additional benchmarks, multiple runs, multiple template approaches) this will be done at the end.
>
> ### Rerun with multiple runs
> As part of running more benchmarks we will conduct at least 5 independent runs (with set random seeds) to obtain mean and standard deviation statistics, to provide more confidence in our results. Our current preliminary results have a total of 3 independent runs.
>
> ### Ablation on template sensitivity
> Yes, we have added a section in the appendix for an ablation study on three approaches to the created templates: one with the manually created templates, one with a generic template, and one with a “meta-prompt” for a dynamically generated template. We run the ablation study on the existing and additional benchmarks to determine template sensitivity and generalisability. The preliminary results show that intrinsic memory without a structured template performs on par with or better than both LLM-generated and manually created templates. This suggests that structured memory templates may not be necessary, but we will finalise our evaluation of these conclusions once all results are finalised and update our paper structure.

---

### Official Review · Reviewer_dG8t · 2025-11-01

**Soundness:** 3
**Presentation:** 3
**Contribution:** 2
**Rating:** 4
**Confidence:** 3

**Summary:**

This paper introduces Intrinsic Memory Agents, a multi-agent LLM framework with structured agent-specific memories (intrinsically updated) to solve context window limits. It outperforms peers by 38.6% on PDDL, excels in data pipeline metrics, and maintains top token efficiency. Its structured memory templates align with agent roles, fixing gaps of methods like RAG that lack role-specific consistency. In data pipeline tasks, it offers specific tools (e.g., AWS Kinesis) and trade-offs, outshining baselines in key quality aspects.

**Strengths:**

- **Novel and Relevant Methodology:** The authors do a good job of identifying a real-world limitation of multi-agent systems: how homogeneous memory causes agents to lose their specialized perspectives. Their proposed "Intrinsic Memory" with "Structured Templates" is a novel and well-thought-out solution to this problem. This approach seems to directly target the core challenges of role adherence and perspective inconsistency.
- **Rigorous Case Study Evaluation:** The data pipeline design case study is excellent. The authors used a complex, realistic task involving eight specialized agents. Critically, they based the evaluation on 10 independent runs. Their use of an LLM-as-a-Judge, combined with a Wilcoxon rank-sum test (as shown by the p-values in Table 2), is methodologically rigorous and makes the results convincing.
- **High Reproducibility:** I was impressed by the commitment to reproducibility. The paper provides extensive details in Section 8 and the appendix. This includes the key algorithms (Alg. 1, 2), the exact memory update prompt (Fig. 4), the task prompt (Fig. 5), and the evaluation prompt (Fig. 6). This level of transparency is commendable.

**Weaknesses:**

- **Manual Template Engineering and Limited Generality:** The method's biggest weakness is its reliance on "manually created," task-specific templates. The authors acknowledge this in Section 6. This reliance on expert human effort means the framework can't be easily generalized to new tasks or different agent roles, which severely limits its practical, out-of-the-box utility.
- **Methodological Flaw in PDDL Benchmark:** I am not convinced by the conclusions from the PDDL benchmark (Section 4). The results in Table 1 are based on "a single run". Given the high variance of LLM agent systems, a single run is not sufficient to draw any robust conclusions.
- **Significant Token Overhead:** This approach introduces a non-trivial cost. The IMA system used 32% more tokens in the case study (Table 2). This extra cost seems to come from two places: 1) the extra LLM call (f_memory_update) needed for every single memory update, and 2) the longer input context (C_n,m) that each agent receives, as it now includes the structured memory.
- **Low Absolute Scores on Key Metrics:** It's important to note that while IMA performed better in *relative* terms, its *absolute* scores for "Usability" (3.67/10) and "Documentation" (3.56/10) are still quite low (Table 2). This suggests that even with a better memory system, the underlying base model (Llama-3.2-3b) struggles to produce documentation that is truly usable for engineers.

**Questions:**

1. **On template generality:** Regarding the main weakness (manual templates), have you explored any ways to automate this? For instance, could a "meta-prompt" be used to generate the JSON structure for an agent (MT_n) just from its role description (R_n)?
2. **On PDDL robustness:** Could you please clarify the PDDL benchmark results (Table 1)? Specifically, can you provide the mean and standard deviation over multiple independent runs (e.g., 5-10)? Without that, it's difficult to assess the confidence of those findings.
3. **On token overhead:** Could you provide a more detailed breakdown of the 32% token overhead? I'm curious how much of that cost is from the extra f_memory_update call (Eq. 3) versus the cost of simply having a longer context for the main agent (L_n, Eq. 2)?
4. **On performance bottlenecks:** Are the low absolute scores for "Usability" and "Documentation" a fundamental limitation of the Llama-3.2-3b model, or is it a limitation of the IMA framework? In other words, if you swapped in a more powerful model (like GPT-4o), would you expect those scores to jump significantly?

---

> ### Author Response · Authors · 2025-11-20
>
> Thank you for your comments, we respond to each of them below.
>
> ### Template generality
> We agree there is limited generality to the use of manually created templates, requiring users to hand-craft a new template when solving new problems. As part of the update to the paper, we conduct an ablation study with three approaches to the use of memory templates: manually created templates as the baseline method, a generic template that can be used across problems without any changes, and finally a “meta-prompt” for dynamically generating templates, prompting each agent at the start of the task to create a template for itself based on its task and role description. The preliminary results show that intrinsic memory without a structured template performs on par with or better than both LLM-generated and manually created templates. This suggests that structured memory templates may not be necessary, but we will finalise our evaluation of these conclusions once all results are finalised and update our paper structure.
>
> ### PDDL robustness
> The current PDDL benchmarks only use a single run, however, as part of the ablation study and improved benchmarking we are conducting more benchmarks (FEVER and ALFWorld datasets) with more runs (at least 5 independent runs, all with set random seeds) to provide mean and standard deviation over the runs. We hope this provides more confidence on the findings.
>
> ### Token overhead
> While our priority is to run additional benchmarks and to complete our ablation study, we will investigate the spread of token usage on the final results, and determine the cost of the memory update call compared to the longer context of the main agent.
>
> ### Performance bottlenecks
> We will run the data pipeline case study on a more powerful model to determine if this affects the absolute scores of the pipeline design. However, we note that the purpose and motivation of our Intrinsic Memory system is to address the limited context and performance of small language models, which becomes exacerbated in a multi-agent system. A larger powerful model has fewer limitations processing the large contexts generated by a multi-agent system conversation.

---

### Author Response · Authors · 2025-11-20
**Overall response to reviewers (Updated December 3rd)**

We thank the reviewers for their constructive feedback. Their main concerns focused on the need for stronger ablations and benchmarking of the intrinsic memory mechanism, as well as clearer isolation of the contribution of structured memory templates to task performance. We summarise our key improvements below:

### 1. Expanded ablation study (Appendix B) with improved benchmarking
- We compare three memory template conditions: a generic non-structured template, an LLM-generated template, and a manually designed template.
- For each condition, we run at least five repetitions across three benchmarks (FEVER, PDDL, ALFWorld) and two models (Mistral 7B, Gemma 3-12B).

Results: Our results show that intrinsic memory without a structured template performs on par with or better than both LLM-generated and manually created templates:
- Mistral:7b([PDDL: 0.066 for no template, 0.069 for LLM-generated, 0.063 for manual template]
- Mistral:7b([FEVER: 0.379 for no template, 0.3009 for LLM-generated, 0.11 for manual template])
- Mistral:7b([ALFWorld: 0.029 for no template, 0.014 for LLM-generated, 0.15 for manual template])
- Gemma3:12b([PDDL: 0.26 for no template, 0.25 for LLM-generated, 0.25 for manual template]
- Gemma3:12b([FEVER: 0.65 for no template, 0.64 for LLM-generated, 0.64 for manual template])
- Gemma3:12b([ALFWorld: 0.048 for no template, 0.045 for LLM-generated, 0.025 for manual template])

These results demonstrate that a generic memory update template is better, and that a manual template approach can be sensitive to a poorly designed template.

### 2. Updated benchmarking section
- Based on the ablation, we will update Table 1 to report PDDL, FEVER, and ALFWorld results using intrinsic memory updates based on the best performing template approach from the ablation study.


### 3. Improved data pipeline case study
- We have updated the data pipeline case study with our results for the intrinsic memory update mechanism with a generic template.

We again thank the reviewers. Their comments directly led to a clearer, more streamlined contribution and improved empirical results. Detailed responses to individual comments follow below under each review.

# December 3rd Update:
Our ablation results demonstrate that the intrinsic memory update mechanism with a generic memory update template obtains the best performance across all benchmarks, for Mistral:7b and Gemma3:12b, except for PDDL with Mistral, where performance is 5% lower than the LLM-generated template approach. Therefore, we focus our contribution on the intrinsic memory mechanism and a generic memory update template.

We have updated Table 1 to report the PDDL, ALFWorld, and FEVER results with means and standard deviations from multiple independent runs. Our further benchmarking results show that while the Intrinsic Memory approach does not always show the best performance, it shows much high consistency with low standard deviation compared to the best performing memory mechanisms, while still maintaining competitive performance. Bar charts with reported error bars are shown in Figure 2 for this result.

We have also updated our data pipeline case study to use the new generic template approach based on the results of the ablation study, and continued to show the strong performance of the Intrinsic Memory in this case study problem.

---

### Meta-Review · Area_Chair_heG6 · 2026-01-02

**Summary:**

The core concerns of the reviewers focus on two major aspects: insufficient experimental rigor and limited novelty/generality of the technical contribution. Specifically, they criticize three main points: the benchmarking relies on a single run and narrow scope, making the results unreliable; the framework heavily depends on manually designed memory templates, raising doubts about its generalizability and core contribution; and the paper's justification for the technical novelty of the mechanism is insufficient, lacking necessary ablation analyses. These issues collectively indicate that the current evidence is inadequate to support the paper's conclusions, requiring significant strengthening of experimental validation.

**Reviewer Concerns:**

The authors effectively addressed the core empirical concerns by conducting extensive new experiments (adding FEVER and ALFWorld benchmarks, running ≥5 trials per setting, and including ablations across manual, LLM-generated, and generic templates, which surprisingly showed that structured templates aren’t necessary and that the intrinsic memory mechanism itself drives performance). However, some issues remain partially unresolved, such as a detailed breakdown of token overhead, concrete qualitative examples of failure-mode mitigation, and definitive comparisons to state-of-the-art structured memory systems like G-Memory.

**Reviewer Scores:**

Reviewer dG8t and m3MB, who originally gave a score of 4, may remain cautiously supportive, likely holding at 4, with only a modest chance of upgrading to 6 despite the thorough responses. Reviewer uNdj would probably stay at 4, as concerns about technical novelty and failure-mode analysis haven’t been fully alleviated. Reviewer iVU5, initially critical of the paper’s technical contribution and baseline comparisons, might marginally improve from a 2 to a 4 if persuaded by the expanded benchmarks, but is unlikely to go higher unless clear comparisons to state-of-the-art memory systems (e.g., G-Memory) are demonstrated; overall, their stance remains skeptical.

---

### Decision · Program_Chairs · 2026-01-26

Reject